# Application and Validation of an Ecological Quality Index, ISEP, in the Yellow Sea

**Jae-Won Yoo** [1], **Yong-Woo Lee** [2], **Mi-Ra Park** [1], **Chang-Soo Kim** [1], **Sungtae Kim** [1], **Chae-Lin Lee** [1], **Su-Young Jeong** [1], **Dhongil Lim** [3] and **Sung-Yong Oh** [4,*]

1 Korea Institute of Coastal Ecology, Inc., IT302-802, Ssangyong Technopark III, 397, Seokcheon-Ro, Bucheon-Si 14449, Gyeonggi-Do, Republic of Korea
2 Office of Science and Technology, National Marine Fisheries Service, National Oceanic and Atmospheric Administration, 1325, East-West Hwy, Silver Spring, MD 20910, USA
3 Library of Marine Samples, Korea Institute of Ocean Science & Technology, 41, Jangmok 1-Gil, Geoje-Si 53201, Gyeongsangnam-Do, Republic of Korea
4 Marine Bio-Resources Research Unit, Korea Institute of Ocean Science & Technology, Busan-Si 49111, Republic of Korea
* Correspondence: syoh@kiost.ac.kr

**Abstract:** An ecological index of macrobenthic communities is an important tool for assessing the biological quality of habitats and ecosystems. We tested the performance of the inverse function of the Shannon–Wiener evenness proportion (ISEP) with data from the entire west coast of Korea, seasonally sampled from 2006 to 2008. Two validations were performed: (1) examination of the relationship between ISEP and environmental factors and (2) correspondence between the ISEP and the Pearson–Rosenberg (P-R) model for the species-abundance-biomass (SAB) patterns and taxonomical variations. The ISEP was significantly correlated with suspended solids but independent of other natural habitat conditions due to their low to moderate contamination levels. From this, ISEP performed as expected in transitional zones of low salinity and applications across habitats of various sediments. The SAB patterns and taxonomic variations along the ISEP grades showed marked similarities to the P-R model. The only departure was biomass, which increased after the normal status. The increase was interpreted as reflecting a healthier and more mature status. Taxonomic variation patterns and the taxa composition that characterized either poor or healthy status corresponded well with the P-R model. The conformity to the P-R model indicates the capability and potential applicability of the ISEP to other coastal systems.

**Keywords:** biotic indices; environment quality; ISEP; macrobenthic communities; P-R model; Yellow Sea

## 1. Introduction

Coastal ecosystems provide a wide array of ecosystem services (such as food production, pollutant filtering, and detoxification), but they are now over or misused and threatened [1,2]. The degradation of these natural habitats and biodiversity loss occur worldwide and are expected to affect several critical ecosystem services [2]. Korean coastal waters, including benthic habitats, have some of the highest levels of biodiversity worldwide in terms of species richness per area and primary production [3,4]. Nevertheless, severe habitat loss and huge impacts occur due to threats and stressors (e.g., large-scale reclamation in Incheon, Shihwa, and Saemangeum; the Hebei Spirit oil spill in Taean on the west coast; hypoxia in Gamak and Masan Bay on the southern coast; and pelagic organism decline caused by overfishing since the late 1990s). There is an urgent need to improve and initiate a framework for assessing and remediating coastal ecosystems and producing reliable scientific evidence and information for environmental policy and decision-making [5].

Macrobenthic organisms have several advantages for biomonitoring in natural habitats because they (i) are sedentary, thus spatially stable; (ii) are composed of diverse species with different sensitivities and tolerances; and (iii) provide a means to integrate environmental effects over long periods owing to their life-spans ranging from months to years [6–10]. Consequently, indices based on macrobenthic communities have been useful for assessing habitat quality [8]. In contrast to Europe and North America, where some useful indices to measure ecological quality have been developed [7,11,12], such ecological index-related studies and applications have rarely been conducted in Asian countries, especially Korea. Korea is a development-oriented country, and, as described above, the current situation urgently requires ecological tools (e.g., evaluation systems or biological indices) that would help to guide the management and stewardship of natural resources.

Recent studies on biological indices emphasize the importance of using locally validated indices or foreign indices calibrated and tested using local data [13,14]. For this reason, Yoo et al. [10] proposed the inverse function of the Shannon–Wiener evenness proportion (ISEP), a modified index based on the abundance-biomass comparison (ABC [11]), and the Shannon–Wiener evenness proportion (SEP [15]). They calibrated the ISEP grade scale using local data to assess the ecological quality of the Korean coastal waters [10]. Although the previous study examined the validity and performance of the proposed index in various ways, applicability to other areas and further validations through empirical testing are essential steps for any newly proposed index. Another reason for further rigorous validation is to analyze whether the ISEP can be used as a reference for other indices in the same region. Having several well-performing indices per region is a huge benefit for a more reliable assessment because a multiple-indicator approach is better than a single-indicator approach [8,16].

Similar to other ecological indices, the ISEP produces a one-dimensional representation of benthic quality status from multivariate biological data. However, because of the reduction in dimensionality, these indices were not designed to identify the impact sources responsible for environmental degradation in a given region. Rather, these indices should be able to accurately and consistently represent environmental status, in an uncomplicatedly ordered score or categorical unit, across a diversity of locations and habitats [10,17–20]. Thus, to be recognized as a reliable, robust, and universal index, these properties should be tested and proven. Previously, Yoo et al. [10] evaluated the performance of the ISEP over different coastal areas and conditions, including habitat types, stress gradients, sample size variations, and a before-after-control-impact setting. The results demonstrate that the ISEP is an effective and robust index in the studied region.

In this study, we validated the performance of the ISEP from a different standpoint than in our previous work [10]. First, we examined the correlations between environmental factors (including natural controlling factors and potential stressors) and the ISEP grades to understand the stressor–ISEP response relationship. This analysis allowed us to understand whether the index is independent of natural controlling factors, such as sediment grain size and salinity, in assessing environmental status. The independence of an index from natural controlling factors means that the index would not be falsely overwhelmed by only a few natural controlling factors. Instead, a good index should accurately represent the system's status when subjected to stress, as suggested by Diaz et al. [20]. Second, the ISEP was compared to the well-known conceptual model of Pearson and Rosenberg [6] for similarities in describing species-abundance-biomass (SAB) patterns and taxonomic compositions. The conceptual model of Pearson and Rosenberg, often referred to as the P-R model or P-R paradigm, has been widely applied in numerous index-related studies and is used as the standard for simulation studies [7,12,16,21,22]. If similarities were found between SAB patterns along ISEP grades and those of the P-R paradigm (as a standard assay), the suitability of the ISEP would be validated as an environmental index. However, because it is a locally adapted index, the ISEP may show empirical relationships with environmental factors that would differ from the expected outcomes based on the

conceptual P-R paradigm. Therefore, the results of this study should help us understand the range of limitations and applicability of the ISEP.

## 2. Material and Methods

### 2.1. Study Area

The study area, on the west coast of Korea, is located in the eastern part of the Yellow Sea, a marginal sea of the Pacific Ocean surrounded by China and the Korean peninsula. According to NOAA [3] and Heileman and Jiang [23], the Yellow Sea is one of the most productive coastal areas, and its level of primary production is categorized as class 1 (>300 g C/m²/year). The circulation and water mass distributions vary seasonally. Currently, the boundary between the Yellow Sea Bottom Cold Water (YSBCW) and the Tsushima Warm Current in the southern part of the Yellow Sea and the tidal front formed between the YSBCW and the Korean coastal waters in the eastern part are known to establish the distribution of species in the macrobenthic community [24,25]. All sampling stations (*n* = 93) in this study were within 12 nautical miles of land and, thus, within Korean coastal waters (Figure 1) [26].

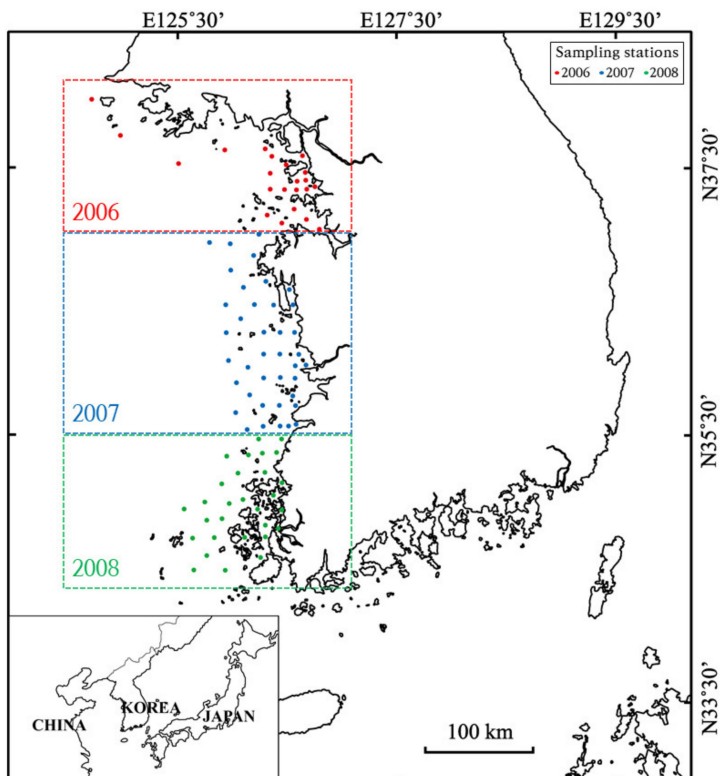

**Figure 1.** Study area on the west coast of Korea. A total of 93 stations were established and sampled seasonally from 2006 to 2008. Circles on the map indicate the sampling station locations.

The average depth of the Yellow Sea is approximately 44 m, with a maximum depth of 103 m in the southeastern part. The tidal amplitude ranges more than 6 m, and, depending on the location, the mean spring tides exceed 9 m [27]. The west coast of Korea is vertically well-mixed because of its shallow depth and strong tidal currents [28]. At spring tide, the flood and ebb currents set north and south with a velocity of about 90–175 and 120–225 cm/s [28]. The surface seawater temperature varies from 2 to 8 °C in winter to 24 to 28 °C in summer. Salinity fluctuates widely because of freshwater inflow, which is concentrated in the rainy and typhoon seasons from early summer to early fall [29]. During winter, salinity ranges from 31 to 34 psu throughout the water column, However, in the rainy season, the surface seawater salinity declines to 30–32 psu, which is lower than in other Korean coastal areas.

The west coast of Korea is a typical ria coast, where the coastline is complex and surrounded by islands. The average dissolved oxygen (DO) remained over 8 mg/L except in some inner bay areas during the summer. In general, the study area is uncontaminated by nutrients or heavy metals. However, some of the closed/semi-closed bays and eutrophic estuaries, such as the Incheon coast, artificial Lake Shihwa in Gyeonggi Bay, and Chonsu Bay, appeared to be contaminated by heavy metals and nutrients and experienced hypoxia [30–34]. In particular, Lake Shihwa, where loss of ecosystem services occurred, resulting from azoic or extremely poor benthic assemblages, maintained for more than a decade after the dyke construction in the mid-1990s until the tidal power plant operation in 2011, became a lesson of unwise ecology in the Korea [10,31].

The west coast is also characterized by the presence of well-developed macrotidal flats [27]. Although coastal wetlands have been globally known to perform critical functions such as provision of nursery habitats and clean water, as well as coastal protection from flooding and storm events [2], the history of reclamation is long in this area, and large-scale reclamation projects (e.g., New Songdo International City, Incheon and Saemangeum Complex, Gunsan, etc.) have been carried out until recently [10]. There is concern that the coastal wetland loss due to reclamation is playing a critical role behind the substantial changes in biodiversity and the decline of water quality, fisheries, and other beneficial ecosystem services [2,10,23].

A peculiar coastal mud belt (southeastern Yellow Sea mud, SEYSM) occurs along the southwestern coast of Korea. It is an enormous Holocene mud deposit approximately 250 km long, 20–50 km wide, and up to 60 m thick, occupying an area of more than 8100 km$^2$ [35,36]. The sediments of the SEYSM are mainly composed of fine-grained silt and clay particles with less than 10% sand. The northern half of the mud belt consists of unconsolidated Holocene muds, whereas the southern half is covered by older semi-consolidated muds [37]. Such a coastal mud belt is associated with an abundant material supply from Korean and Chinese rivers and strong tidal currents [36,38]. Because the seafloor muds of the SEYSM deposit respond vigorously to the combined action of waves and tidal currents, a turbid water plume always occurs in this area. In the center of the plume, suspended solids (SS) exceed 200 mg/L, especially during winter, and abruptly decrease offshore from >20 mg/L to <10 mg/L to the plume's outer (western) front [36,39].

*2.2. Field Sampling and Laboratory Analysis*

Ninety-three stations were established along the western coast of the Korea (Figure 1). Three subdivisions of the coast were consecutively conducted from north to south over three years (2006–2008). Macrofaunal samples were collected at seasonal intervals every year in the subdivision under study using a Smith–McIntyre or Van Veen grab (0.1 m$^2$). Three replicates were performed at each station. Each sediment sample was sieved on a board in a 1 mm round mesh sieve, and the resultant residues were preserved immediately in 10% neutral buffered formalin. In the laboratory, faunal samples were sorted and identified at the species level, if possible, and counted under a dissecting stereomicroscope. Biomass was measured as the wet weight (g) using an electronic balance. The abundance and biomass were converted to per-unit area (m$^2$) values.

Water column environmental data and bathymetry data were obtained onboard using automatic monitoring devices (CTD, Seabird-911, Sea-bird Scientific, Bellevue, WA, USA; Multiparameter Sonde, YSI-6600, YSI Incorporated, Yellow Springs, OH, USA)) and SONAR. Water samples from the surface and bottom waters were collected using a rosette sampler. Water column DO was determined using the Azide–Winkler titration method, and averages were estimated using 2–4 replicates at each station. Particulate organic carbon (POC) and particulate nitrogen (PN) were estimated using an elemental analyzer (Flash EA 1112, Thermo Fisher Scientific, Waltham, MA, USA) after filtering the seawater through a GF/F filter (Cytiva, Marlborough, MA, USA) and exposing it to 10% HCl fumes. Laboratory analysis was performed in duplicate or triplicate to determine suspended solids (SS), POC, and PN, according to MLTM [40]. Standard material (L-Cistina, Thermo Fisher Scientific,

Waltham, MA, USA) was used to test the reliability of the measurements. The recovery ratio was between 92 and 108% for all three years. Chlorophyll-*a* concentration was measured using the fluorometric method (pigment extraction in acetone and fluorescence detection using a Wet Lab fluorometer, General Oceanics, Miami, FL, USA).

Sediment samples were obtained from the same stations where macrofaunas were sampled. Surface sediments (0–2 cm depth) were taken to the laboratory, and grain size analysis was performed using Ingram's wet/dry sieving and pipetting methods [41] and the granulometric equations of Folk and Ward [42]. For elemental analysis, 0.2 g of each powdered sediment sample was dissolved in 6 mL of HF and $HClO_4$ (2:1 mixture). This was then evaporated to dryness, cooled, and dissolved in 20 mL of 10% $HNO_3$. This solution was analyzed for trace elements using inductively coupled plasma atomic emission spectroscopy and mass spectroscopy (ICP-AES, OPTIMA 3300R and ICP-MS, ELAN 5000, PerkinElmer Inc., Waltham MA, USA). A standard reference material (USGS MAG-1) was analyzed in addition to the sample sets to provide control for the analytical precision and accuracy. The results showed that the differences between the determined and certified values were generally less than 10% (except for Cd), indicating satisfactory recovery. To understand the intensity of anthropogenic contamination by trace elements, we used the enrichment factor (FE; [43]), which is expressed as follows:

$$EF = (Metal/RE)_{soil}/(Metal/RE)_{background}$$

where RE is the reference element. We used aluminum as RE because it is known as a conservative element and a major constituent of clay minerals [43]. The background values were obtained from the averages in the Earth's crust, as suggested by Taylor [44]. EF < 2 indicates deficiency to minimal enrichment, and EF > 2 indicates that a significant portion was delivered from non-natural sources; 2 < EF < 5 indicates moderate enrichment; 5 < EF < 20 indicates significant enrichment; 20 < EF < 40 indicates very high enrichment; and EF > 40 indicates extremely high enrichment [43].

The percentages of total carbon (TC) and nitrogen (TN) in the sediment were measured using a Carlo Erba elemental analyzer 1108 (CE Instruments Ltd., Wigan, UK). The percentage of total inorganic carbon (TIC) was measured using a $CO_2$ coulometer (Model CM5014; UIC Inc., Joliet, IL, USA). The analysis was accurate, with an analytical error of 5%. The percentage of total organic carbon (TOC) was estimated as the difference between the percentage of total carbon and the percentage of total inorganic carbon. Assuming that all assessed inorganic carbon occurred as calcium carbonate, the percentage of $CaCO_3$ was calculated as a weight percentage from the percent TIC using a conversion factor of 8.33 g $CaCO_3$ g/C. Analytical methods for environmental data are described in the MLTM [26].

### 2.3. Index Application and Validation

The ISEP index [10] is a modification of the Shannon–Wiener evenness proportion (SEP; [15]) and is calibrated for Korean waters to establish a local grade scale. Because of its relationship with the SEP, the ISEP can be regarded as a derivative of the abundance-biomass comparison (ABC) method [11]. The formula for calculating the ISEP is as follows:

$$ISEP = \log_{10}(1/SEP + 1)$$

$$SEP = E(biomass)/E(abundance) = H'(biomass)/H'(abundance)$$

where *E* refers to evenness and *H'* is the Shannon–Wiener diversity index [45]. A higher index value intuitively indicates that the macrofaunal sample from a habitat retains conservative species with large body sizes and long lifespans (*K*-strategists). Therefore, high ISEP values are expected for a healthy environment, and low ISEP values are often found in stressed environments [10]. Seven grades were suggested based on percentiles of the ISEP distribution in Korean coastal waters based on quintiles, namely 20, 40, 60, and 80th, estimated by Yoo et al. [10], to which we added 10 and 90th percentiles to discriminate

habitats of extremely bad or good status. Bootstrapping was used to obtain robust ISEP grade scores for each percentile band (Table 1).

**Table 1.** ISEP grading criteria based on percentiles estimated from 5000 bootstrap samples (modified from Yoo et al. [10]).

| ISEP Grade | VII | VI | V | IV | III | II | I |
|---|---|---|---|---|---|---|---|
| Percentile | ≤10 | 10< | 20< | 40< | 60< | 80< | 90< |
| Average | | 0.144 | 0.291 | 0.359 | 0.426 | 0.61 | 0.912 |
| SE | | 0.047 | 0.007 | 0.003 | 0.005 | 0.014 | 0.043 |
| 2 SE | | 0.094 | 0.013 | 0.005 | 0.011 | 0.028 | 0.085 |

The ISEP index was validated in three ways. First, the index was applied to the benthic macrofaunal community on the west coast of Korea by calculating an ISEP score for each macrofaunal replicate sample (0.1 m$^2$) and then averaged to estimate an ISEP grade for each station. To evaluate the performance of the index, Pearson's correlation analyses were carried out to understand the relationships between the ISEP grades and the environmental factors of the sediment and bottom water of the west coast of Korea. We expected significant correlations of the ISEP with potential stressors (e.g., organic matter content, suspended solids, heavy metals) but no significant correlation with natural factors (e.g., depth, temperature salinity, sediment mean grain size).

Second, we built graphs of the variations in species-abundance-biomass (SAB) of the macrofaunal community along the ISEP grades on the west coast of the Korea, and those along the stress gradient of the P-R model [6], and analyzed the agreement between them. This validation was based on two assertions: (1) the ISEP grade would effectively reflect the stress gradients along the west coast of Korea, and (2) the similarity between the observed patterns in SAB variation across the ISEP grades and the expected patterns from the P-R model indicates the conformity of the ISEP to the P-R model, providing evidence that the ISEP is an effective ecological indicator.

For another perspective on the pattern of variation, SAB variation for major taxonomic groups and dominant species was examined across ISEP grades and compared with successive taxonomical changes originally proposed in the P-R model. For the dominant species, the top 20 species were selected separately by total abundance and biomass, and 8 species were chosen based on their frequency of occurrence. From this, we expected that the typical responses of taxa and $K/r$-strategists could be observed along the ISEP grades reflecting stress gradients. For the species' life history strategies, we referred to the related references [6,7].

### 2.4. Measuring Agreements in SAB Patterns

Rosenberg et al. [46] classified the benthic habitat quality (BHQ) and the benthic quality index (BQI) into five ecological statuses (high, good, moderate, poor, and bad). We used these categories as compatible references and to validate the objectivity and reliability of the ISEP index. Our classification was based on the characteristic phases in the P-R model (e.g., the peak of opportunists, ecotone point, and transition zone) and the normative definitions (e.g., good: the level of diversity and abundance slightly outside normal; poor: major alterations or substantial deviations from normal) of the European Union (EU) The Water Framework Directive (WFD) Annex V relating to benthic invertebrates is displayed in Figure 2.

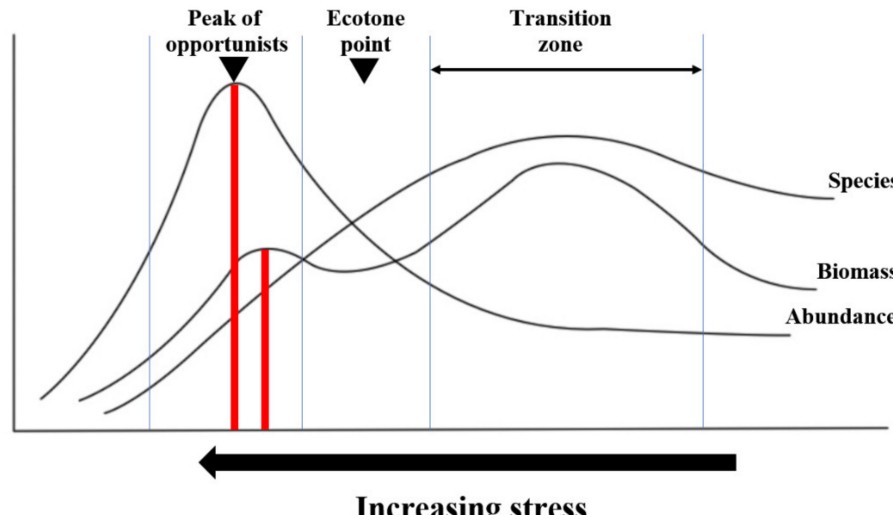

**Figure 2.** P-R model [6] redrawn for the classification of the five ecological statuses based on the successional characteristics and the normative definitions of WFD Annex V relating to benthic invertebrates. The red bars are benchmarks for measuring the relative magnitudes of SAB to estimate the degree of agreement between the SAB pattern along seven ISEP grades and those in P-R model.

To analyze the conformity of the ISEP to the P-R model, we assessed the agreement between SAB patterns along the ISEP grades and the status categories in the P-R model. The data used for the comparison were SAB averages observed on the west coast of Korea, standardized between −1 and 1, for the former. For the latter, we directly measured the unitless relative values of SAB for each status category from the graph of the P-R model (see the red benchmark bars used for the measurements in Figure 3). The estimated SAB values are listed in Table 2.

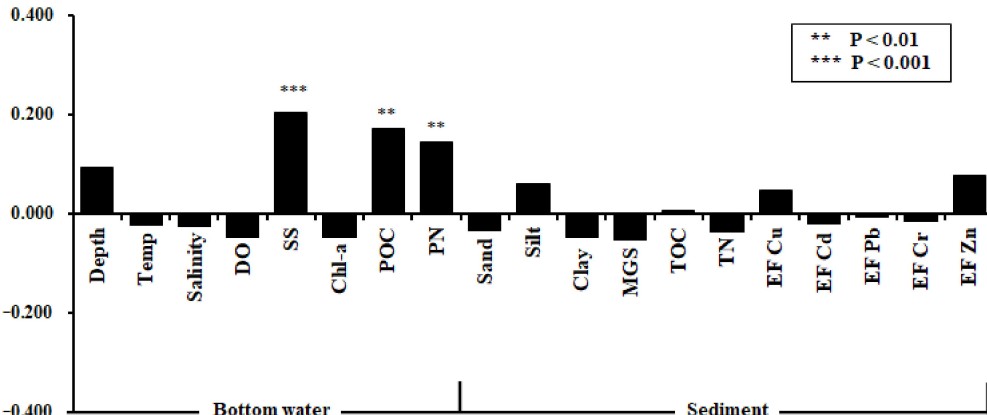

**Figure 3.** Correlations between ISEP grades and environmental factors along the west coast of Korea.

**Table 2.** Unitless value of SAB estimated from P-R model to use as a reference to validate the ISEP index.

| | P-R Model | | |
|---|---|---|---|
| Ecological Status | Abundance | Biomass | Species |
| Bad | 0 | 0 | 0 |
| Poor | 7.6 | 4 | 2.4 |
| Moderate | 4.6 | 3.5 | 4.5 |
| Good | 2.2 | 5.8 | 6.4 |
| High | 2.1 | 3 | 5 |

As a statistical tool for shape correspondence analysis between SAB patterns along ISEP grades and the categories along the stress gradient in the P-R model, we chose the correlation-based technique suggested by Clarke and Ainsworth [47], which measures the matching degree of two different biological and environmental MDS ordination configurations by estimating the rank correlation between triangular similarity matrices underlying both ordinations. We estimated the Spearman rank correlation and significance between the standardized scores of the SAB averages along the ISEP grades and the relative SAB values along the categories in the P-R model (Table 2).

The ISEP grades were divided into seven categories (I to VII), while the categories in the P-R model were divided into five categories, as mentioned above. Considering the different numbers of grades/categories, two different approaches were used to match the SAB patterns. First, to compare the ISEP grades I and II, we stretched two more categories over the high-quality status of the P-R model by allocating the same SAB values. Second, to compare the five categories in the P-R model, we cut the ISEP grades I and II. The premise of these comparisons was that ISEP grade III corresponded to the high-quality status in the P-R model. It was based on (1) the SEP, an original version of the ISEP, validated using Pearson's [48] Loch Linnhe and Loch Eil data [15], (2) the quality status of the SEP scores suggested by Wilson and Jeffrey [49], and (3) the one-to-one relationship between the SEP and ISEP [10].

To assess the SAB pattern agreement between the ISEP and P-R models, we first took an individual parameter level approach for each species number, abundance, and biomass, and then an integrated level approach, which was used to assess overall agreement. We compared and analyzed whether possible different similarities or results existed among them. Standardized averages of SAB were obtained from the three subdivided regions of the west coast (i.e., north, middle, and south). Tests on the overall mean values of SAB on the west coast were excluded because of the low statistical power stemming from the limited sample size (that is, max. $n = 5$ or 7 for different grades).

## 3. Results

### 3.1. ISEP Variation with Environmental Factors

The biological and environmental data used in the correlation analyses are summarized in Table 3. The variation in biological parameters reflected the great differences among sites from extremely poor (almost azoic) to diverse (141 species in a station with a 0.3 m$^2$ surface sediment) or productive status (more than 2 kg of wet weight in 1 m$^2$). A wide range of habitats was included in this study in terms of salinity (14–34 psu) and sediment types (mean grain size from $-0.8$ to 8.2 $\Phi$). Some of the habitats seemed to be stressful environments, such as having low oxygen (DO < 4 ppm), high turbidity (SS > 800 mg/L), and eutrophication (Chl-a > 10 ug/L). The EF averages of trace elements ranged from 0.24 (copper) to 2.75 (lead), and most of the elements corresponded to the deficiency level. The maximum EF of lead was 7.82, a level of significant enrichment, but the upper bound of two standard deviations of EF was 4.63 and fell in the category of moderate enrichment.

Correlation analyses were performed to understand the types of stressors related to the ISEP grade variation along the west coast of Korea (Figure 3). Significant positive correlations were observed between the ISEP grades and a few environmental factors (bottom water POC, PN, and SS). A positive correlation indicated that, in this case, habitats with lower levels of bottom water POC, PN, and turbidity corresponded with lower ISEP grade values (i.e., higher habitat quality) and vice versa. One important aspect that should be noted in the correlation analyses is that the ISEP had no significant relationship with natural controlling factors (e.g., sediment type, depth, bottom water temperature, and salinity) or potential pollutants with low concentrations (e.g., sediment TOC and trace elements).

**Table 3.** Biological (macrofaunal community) and environmental (bottom water and sediment) data used in correlation analysis.

**Biological Data (Macrobenthic Community)**

|  | Species (0.3 m$^2$) | Abundance (m$^2$) | Biomass (m$^2$) | ISEP Score Mean (0.1 m$^2$) | ISEP Grade |
|---|---|---|---|---|---|
| Min | 1 | 3 | 0.08 | 0.000 | 1 |
| Max | 141 | 36,791 | 2234.44 | 1.679 | 7 |
| Average | 47 | 1457 | 92.54 | 0.499 | 3 |
| SD | 28 | 2965 | 215.22 | 0.208 | 1 |

**Environmental data (bottom water)**

|  | Depth (m) | Temperature (°C) | Salinity (psu) | Dissolved oxygen (mg/L) | Suspended solids (mg/L) | Chlorophyll-a (μg/L) | POC (uM) | PN (uM) |
|---|---|---|---|---|---|---|---|---|
| Min | 3.0 | 2.1 | 14.7 | 3.5 | 0.7 | 0.0 | 9.4 | 1.3 |
| Max | 65.0 | 28.7 | 34.0 | 13.1 | 845.1 | 15.0 | 788.0 | 86.0 |
| Average | 23.4 | 13.5 | 31.6 | 8.6 | 56.2 | 2.0 | 78.3 | 9.6 |
| SD | 14.9 | 6.5 | 1.9 | 1.7 | 103.5 | 2.0 | 93.8 | 10.0 |

**Environmental data (sediment)**

|  | Sand (%) | Silt (%) | Clay (%) | MGS (Φ) | TOC (%) | TN (%) | EF Cu | EF Cd | EF Pb | EF Cr | EF Zn |
|---|---|---|---|---|---|---|---|---|---|---|---|
| Min | 0.0 | 0.0 | 0.0 | −0.79 | 0.00 | 0.00 | 0.04 | 0.00 | 1.26 | 0.09 | 0.24 |
| Max | 100.0 | 85.0 | 56.2 | 8.21 | 2.32 | 0.20 | 1.30 | 3.29 | 7.82 | 1.23 | 3.75 |
| Average | 59.6 | 27.1 | 11.7 | 3.86 | 0.30 | 0.04 | 0.24 | 0.53 | 2.75 | 0.58 | 1.01 |
| SD | 36.2 | 25.5 | 14.1 | 2.06 | 0.29 | 0.04 | 0.13 | 0.59 | 0.94 | 0.21 | 0.43 |

### 3.2. Species-Abundance-Biomass Variation along ISEP Grades

SAB patterns along ISEP grades across all samples in the data differed among the parameters but were generally non-linear (Figure 4). Log-transformed biomass showed an overall increasing trend from ISEP grade V to grade I, while abundance showed an overall decreasing pattern from grades VI to I, after a steep increase from VII to VI for both cases. The species number reached a maximum at grade IV, dipped, and stabilized from grades III to I. This pattern of SAB, as shown in Figure 4, closely conforms to the P-R model of Pearson and Rosenberg [6]. The characteristic zones of the grossly polluted zone (GPZ) with no macrofauna, peak of opportunists (PO), transitory zone (TZ, referred to as TR in the P-R model paper), and the normal zone coincide with ISEP grades VII, VI, IV-V, and I-III, respectively.

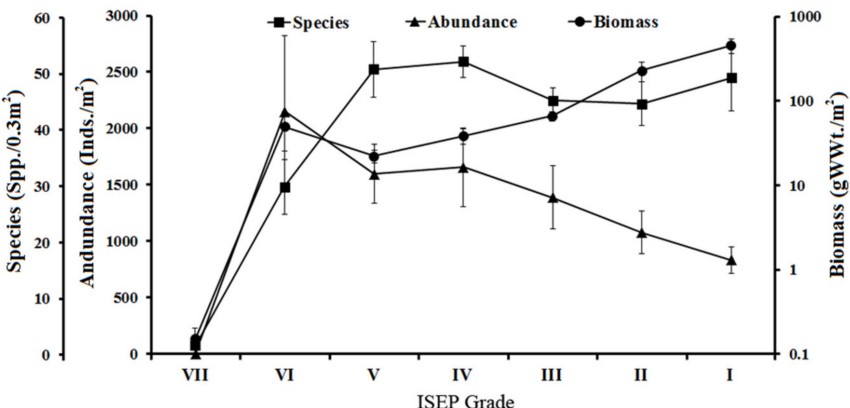

**Figure 4.** Overall pattern of SAB (species-abundance-biomass) variation in the macrofaunal community along ISEP grades observed along the west coast of Korea. ISEP grades are shown on the *x*-axis from low quality (grade VII) to high quality (grade I), from left to right, to allow comparison with the P-R model. Note that biomass is presented on a logarithmic scale. The points are averages, and the vertical bars represent the standard errors.

### 3.3. Matching of SAB between ISEP and the P-R Model

The results of the shape correspondence analysis of species, abundance, and biomass (SAB) patterns between the ISEP grades and the status categories in the P-R model are presented in Figure 5. Most of the comparisons showed strong and significant correlations larger than 0.6 or 0.7, indicating high matching degrees at the individual and integrated levels. Similar correlations were observed between the seven and five ISEP grade comparisons (Figure 5a,b). Biomass was the only insignificant parameter and showed contrasting correlations between the seven- and five-grade comparisons (r = −0.07 vs. 0.32). Based on the higher correlations in the SA combination than in SAB, and the higher correlations in SAB and B of five grades than seven, the biomass pattern in ISEP grades I and II was thought to be the source of the difference in the matching degree.

### 3.4. Patterns of Major Taxa and Species over ISEP Grades

The patterns of SAB by taxonomic group suggested which taxonomic groups contributed to the SAB pattern along the ISEP grades (Figure 6). The overall patterns in species number and abundance along the ISEP grades appear to have been mainly driven by annelids. Other taxonomic groups showed similar trends and patterns with much lower magnitudes (e.g., arthropods and mollusks) but appeared no longer at grade VII. The pattern of log-transformed biomass was rather complex but appeared to increase toward grade I. The elevated biomass at the grades of normal status (I and II) was mainly due to the contributions of echinoderms, followed by mollusks and arthropods.

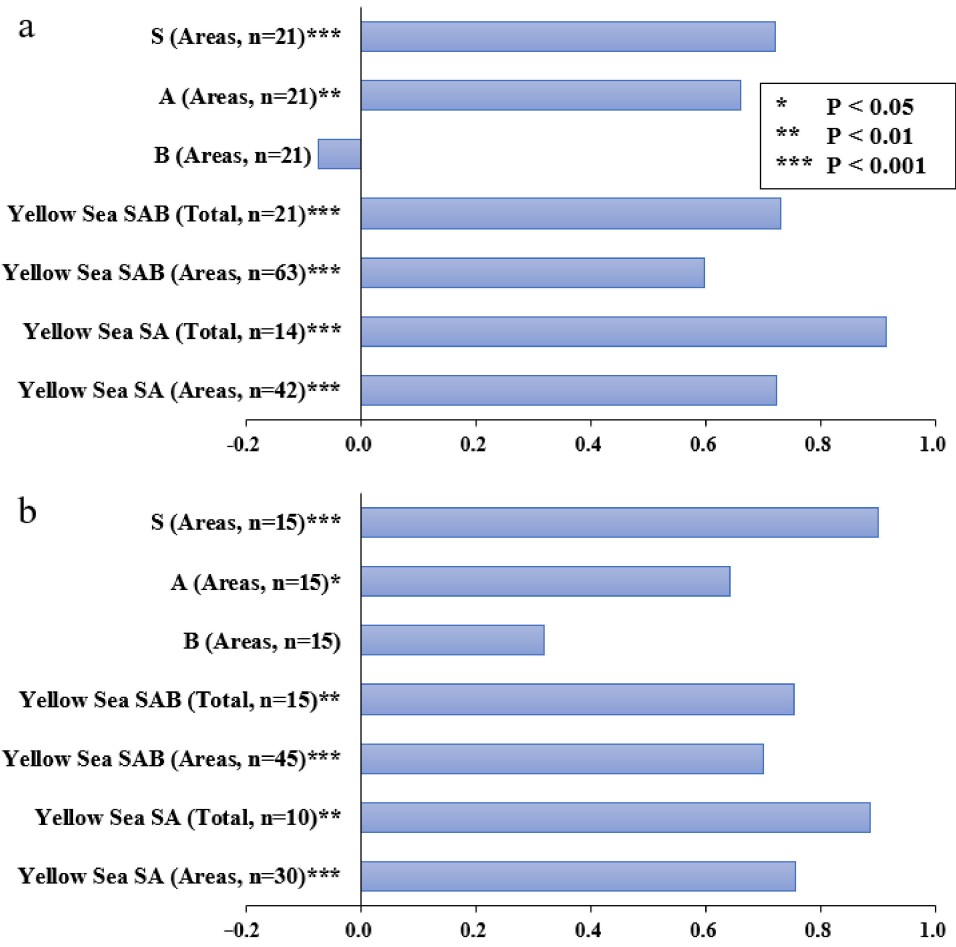

**Figure 5.** Shape correspondence analysis of SAB patterns between ISEP grades and the categories reflecting the stress gradient in P-R model, performed individually (S, A, and B, respectively) and at integrated levels (SAB or SA excluding biomass). Seven ISEP grades (**a**), and five grades (**b**).

Based on the conformity of the ISEP grades to the P-R model, the ISEP grades were categorized into characteristic zones, as explained in the Materials and Methods section. This categorization was used to understand the distribution patterns of the abundance and biomass of the dominant species along characteristic zones (Figures 7 and 8). The minimum total abundance and biomass of the species were 6624 inds. g and 168 g, and the minimum frequencies were 43% and 15%, respectively.

The species with higher abundance were known opportunistic polychaetes (e.g., capitellid, *Mediomastus californiensis*, *Heteromastus filiformis*, *Notomastus latericeus*, cirratulid, and *Tharyx* spp.) and were most abundant in the PO zone (Figure 7). The maximum abundances of ophiuroids, *Amphioplus japonicus*, and lumbrinerid polychaete *Lumbrineris cruzensis* were observed in the TZ. Other polychaetes (e.g., *Sternaspis scutata* and *Nephtys polybranchia*) appeared in the TZ and normal zones. Variations in the abundance of dominant species were not random but monotonic or unimodal.

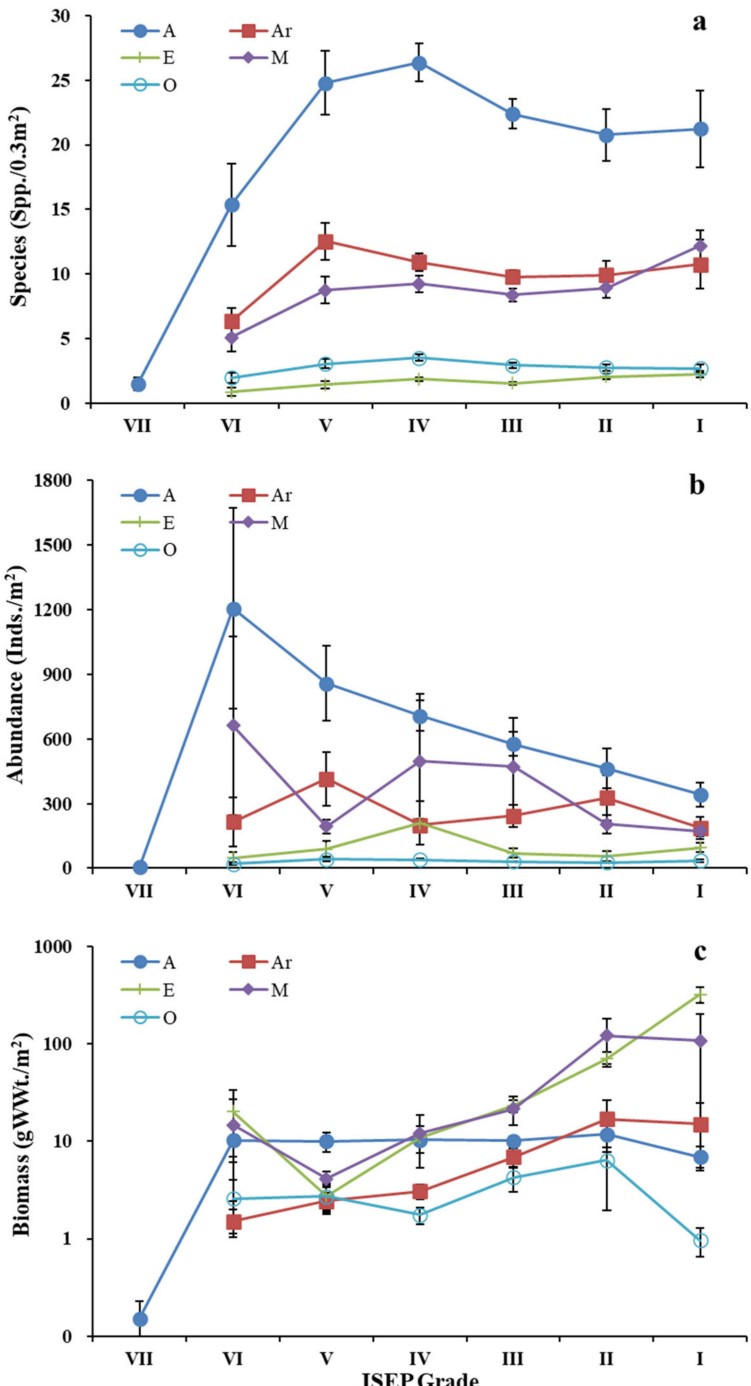

**Figure 6.** SAB (Species, (**a**); Abundance, (**b**); and Biomass, (**c**)) variations of major taxa over ISEP grades. ISEP grades were determined from the data collected along the west coast of Korea. The points are averages, and vertical lines are standard errors. A = annelids, Ar = arthropods, E = echinoderms, M = mollusks, O = others. ISEP grades are shown on the *x*-axis from low quality (grade VII) to high quality (grade I) to allow comparison with the P-R model. Note that biomass is presented on a logarithmic scale.

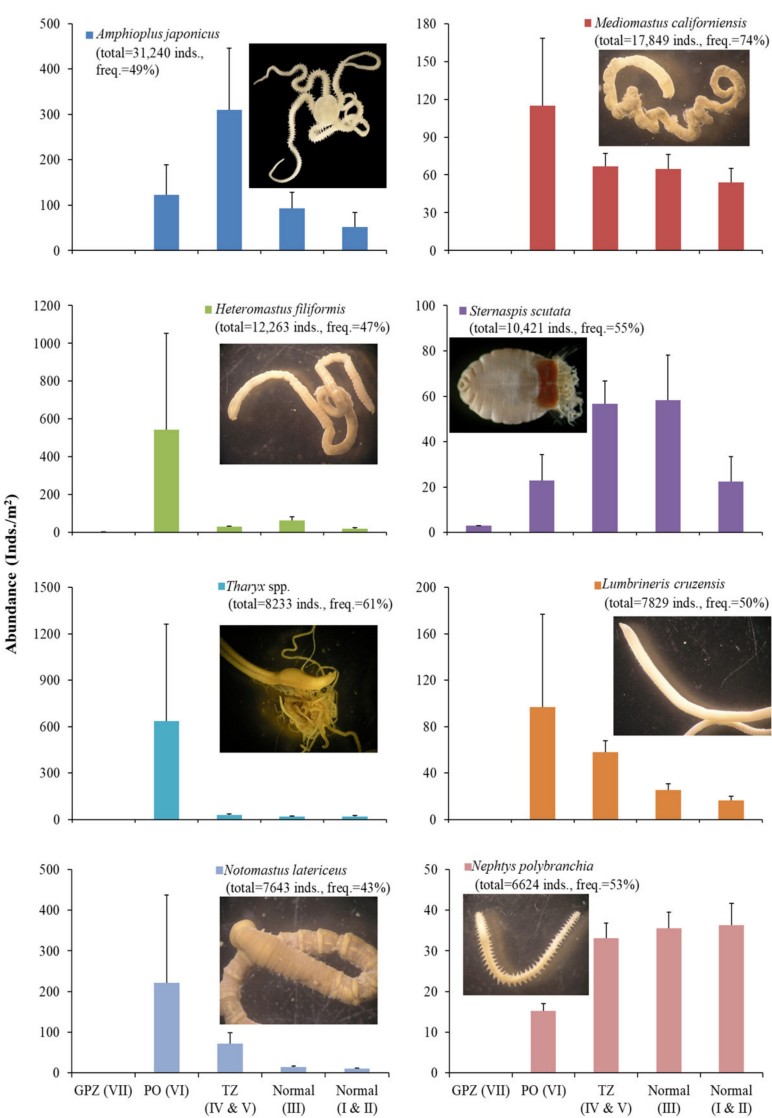

**Figure 7.** Variations in the abundance of dominant species along the characteristic zones. The data are shown for each dominant species of high abundance. ISEP grades are shown on the *x*-axis from low quality (grade VII) to high quality (grade I) to allow the direct comparison with the P-R model. Note that TZ = transitory zone, PO = peak of opportunists, and GPZ = grossly polluted zone.

The patterns of the dominant species, in terms of biomass, were not significantly different in their monotonic or unimodal responses along the characteristic zones. However, the biomass of most species generally increased toward the normal zone (Figure 8). The representative species were spatangoid echinoderm, *Echinocardium cordatum*, pinnotherid crab, *Xenophthalmus pinnotheroides*, polychaetes, *Sternaspis scutata*, *Lumbrineris heteropoda*, tellinid bivalves, *and Nitidotellina hokkaidoensis*. In contrast, the biomass maxima of ophiuroids, *A. japonicus*, polychaetes, *H. filiformis*, and *Glycera chirori* coincided with their abundance peaks and appeared in the TZ/PO zones (Figure 6).

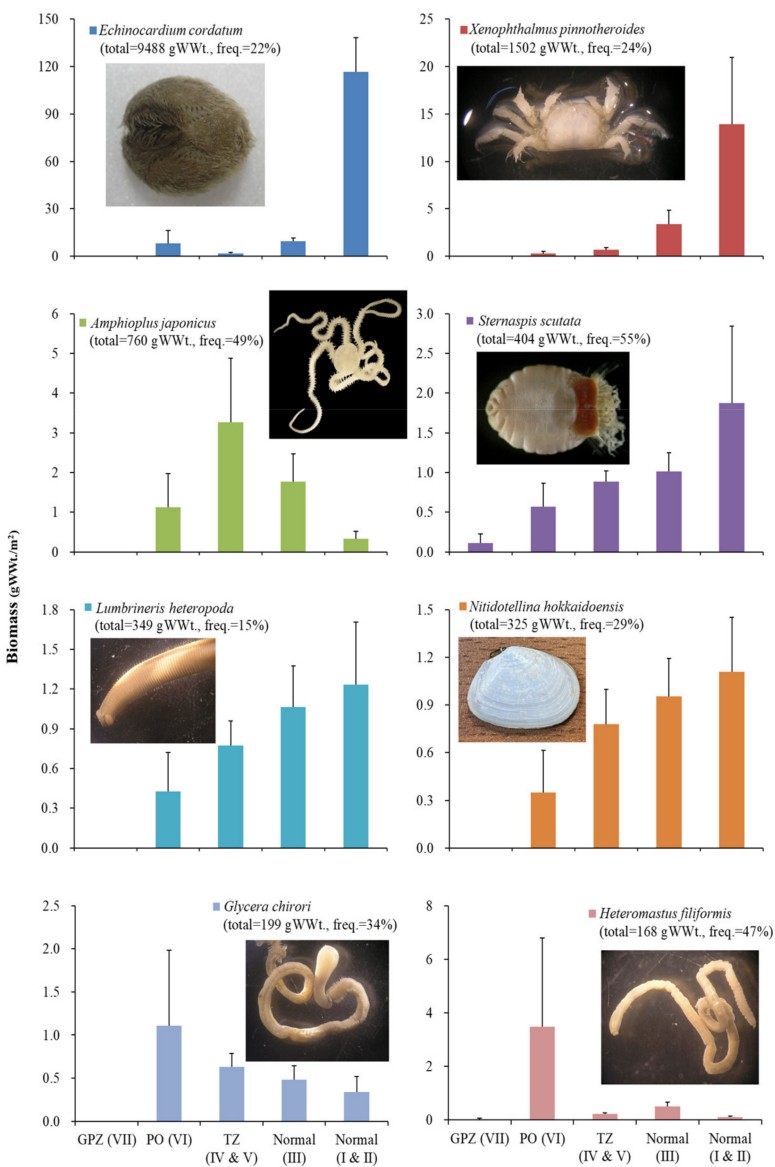

**Figure 8.** Variations in the biomass of dominant species along the characteristic zones. The data are shown for each dominant species of high biomass. ISEP grades are shown on the *x*-axis from low quality (grade VII) to high quality (grade I) to allow the direct comparison with the P-R model. Note that TZ = transitory zone, PO = peak of opportunists, and GPZ = grossly polluted zone.

## 4. Discussion

### 4.1. Relationship with Natural/Artificial Factors

The variability of an ecological quality index to natural factors (e.g., salinity, sediment particle size, and other natural disturbance events) should be discussed as an important part of validation. The ISEP was significantly correlated with bottom water SS, POC, and PN. Drawing a cause-and-effect relationship as to which variable is more critical from correlation alone is difficult. However, based on prior knowledge of the critical role of SS in environmental quality grades [50], SS may be the primary influential factor on habitat quality at the spatiotemporal scale of this study. The apparent correlations between the ISEP and the other two factors (POC and PN) could be mainly due to the variables' close relationship with SS (r = 0.94 (*p* < 0.001) for POC and r = 0.91 (*p* < 0.001) for PN). In a local area adjacent to the southeastern Yellow Sea mud (SEYSM) experiencing

large seasonal variations in SS (6.1–845.1 mg/L [51]), the habitat quality decreased as SS increased, indicated by a correlation between ISEP grade and SS (r = 0.432, *p* = 0.004, *n* = 43).

The ISEP showed no linear relationships with other natural environmental variables, such as depth, temperature, salinity, or sediment textural parameters. An insignificant relationship with salinity needs particular attention. It should be emphasized in the context of the estuarine quality paradox [52] because it may indicate a weakness or bias of established indices when applied to transitional water with variable salinity (0.5–30 psu [8]). Some studies have shown that applying indices developed for coastal waters with higher diversity levels to estuarine regimes with originally low diversity and high abundance [53,54] led to bias due to differences in thresholds [8,9]. The observed insignificant relationships between the ISEP and salinity could be because the ISEP has no functional relationship in a linear fashion with species richness, because the species richness term (ln *s*) in the diversity formula ($E = H'/\ln s$) is canceled out when *E* (biomass) is divided by *E* (abundance) [10]. However, because our salinity data from the west coast showed a skewness towards higher salinities (range = 14.7–34.0 psu, mean = $31.6 \pm 1.9$ psu), compared with other variables, such as depth (3–65 m, $23.4 \pm 14.9$ m) and sediment mean grain size (−0.8–8.2 Φ, $3.9 \pm 2.1$ Φ), further testing is required to understand the ISEP response to salinity.

The lack of a significant correlation between the ISEP and DO, Chl-a, and sediment TOC may be due to a similar reason. For example, the min/max values of bottom-water DO (min. 3.5 mg/L), Chl-a (max. 15 µg/L), and TOC (max. 2.3%) were sufficient to induce local eutrophication stress. However, these variables in our dataset were also skewed on the regional scale (Table 1), with few data points in the lower or higher end of the ranges to exhibit close relationships between the ISEP and these environmental data. In the case of bottom DO showing a slight tendency toward symmetry, the frequency of less than 4 mg/L was <1% (2 out of 362 observations).

Our study showed that the EF values of heavy metals in sediments indicated that the average and maximum values did not exceed the moderate enrichment level (2 < EF < 5), except for Pb. The EF maximum of Pb was observed in the sand bottom in the northern area (MGS, 2.65 Φ and sand content, 97%), and the total concentration was less than 50 ppm, which was close to the NOAA ERL (effects range low, 46.7 ppm; Long and Morgan [55]). For other total heavy metal concentrations, 16% of the samples exceeded the ERL (81 ppm) for Cr (21–108 ppm) in the northern area adjacent to metropolitan cities where more than half of the Korean population lives [56]. In the southern area where the coastal mud belt (southeastern Yellow Sea mud, SEYSM) is located, 50% of the sediment samples were classified as medium silt (6 Φ) to clay (9 Φ), and 48% of the samples exceeded the ERL (20.9 ppm) for Ni (2–36 ppm) [51]. In general, heavy metal concentrations in most sediment samples did not exceed the ERL, and none of the samples exceeded the effects range median (ERM). Although some inner bay and heavily industrialized areas of the west coast have experienced heavy metal pollution [57,58], the sediment samples in this study did not appear to be contaminated. This may explain the lack of a significant relationship between the ISEP grades and heavy metals.

Understanding the response of an ecological quality index to natural factors and its variability is important for assessing its capacity and performance [8,20,59]. An index should not be significantly correlated with these natural controlling factors to be effective and robust. Otherwise, rescaling or calibration of index thresholds should be performed for different habitats [8,20]. It should be noted that results from correlation analyses showed that the ISEP effectively responded to stressors such as turbidity, which was found to be a key natural impact source on the west coast of Korea. In contrast, the ISEP was not correlated with other natural habitat conditions (e.g., salinity and sediment mean grain size), which proves the robustness of the ISEP.

### 4.2. Correspondence with the P-R Paradigm

Pearson and Rosenberg [6] studied macrobenthos SAB curves along the organic pollution gradients along European coasts and in transitional waters (fjord, estuaries, and bays).

They developed the P-R model, which is now widely used as a conceptual framework to describe and understand spatiotemporal variation in the responses to stressors in marine benthic communities [16].

Patterns in the SAB curves over the environmental gradients, represented by the ISEP grades in this study (Figure 4), showed close similarities with those in the P-R model, except for the absence of the secondary biomass peak in TZ (see Section 4.3 below for further discussion). This close correspondence validates the ISEP as an effective index of the environment disturbance gradient. Variations in the major taxa along the ISEP grades are another validation point. As shown in Figure 4, the spiked abundance in PO and the elevated biomass under normal conditions were characterized by particular taxa, as suggested by Pearson and Rosenberg [6] (e.g., annelids in reduced sediments; arthropods and echinoderms in oxidized sediment). In addition, the correspondence between the ISEP and P-R models was supported by the occurrence patterns of the dominant species. Most of the dominant species peaked around PO and TZ. In particular, capitellid and cirratulid polychaetes (e.g., *H. filiformis* and *Tharyx* spp.) occurred at the center of the distribution or at a maximum in the PO. Previous studies have described these species as subsurface-deposit feeders and first- or second-order opportunistic species [6,7,60–64].

While the results from the ISEP applications were quite similar to those with other indices, some findings were very specific to the studied region. Borja et al. [7] placed capitellid polychaetes into ecological group III (i.e., species tolerant to excess organic enrichment, slightly unbalanced situations) out of five groups in AMBI (AZTI's marine biotic index). In Korean coastal waters, however, these species survive in infamously degraded habitats (e.g., virtually azoic habitats typified by severe hypoxia or heavy metal pollution) [30,56,65]. When applied to Korean waters, AMBI placed this species into group IV (second-order opportunistic species occurring from slight to pronounced unbalanced situations) [65].

Species that were not frequently encountered but were found in relatively higher abundances in ISEP grade VI (PO point) were mytilid bivalves, *Arcuatula senhousia* n = 8754, frequency = 5%), spionid polychaetes, *Aonides oxycephala* ($n$ = 3798, frequency = 15%), and *Prionospio japonica* ($n$ = 1428, frequency = 2%). *A. senhousia* is classified as an opportunistic species and is observed in the inner bays where hypoxia occurs [66]. Based on an extensive review of coastal studies on a global scale, Crooks [67] reported that the typical abundance density of *A. senhousia* per unit square meter (individuals/m$^2$) in many parts of the world ranged between 5000 and 10,000. We observed a much higher density of this species (10,000 to >20,000 inds./m$^2$) in Lake Shihwa, an artificial lake on the west coast of the Korea, where severe hypoxia has persisted for a while (unpublished data). The polychaetes *A. oxycephala* and *Prionospio* spp. are organic load indicators or second-order opportunistic species [7,68]. The former species is possibly a physical disturbance indicator because of its higher densities (maximum density >12,000 inds. m$^2$) observed in the bottom trawl permitted area on the southern coast of the Korea [69].

However, not all species with higher abundance in ISEP grade VI were indicator species. A typical example is the hen clam *Mactra chinensis* ($n$ = 7997 individuals, frequency = 9%), which occasionally has high densities similar to other opportunistic mactrid bivalves (i.e., *Mulina*). However, this important commercial species is currently not correlated with environmental disturbance. While one of the grade VI samples had the highest abundance of *M. chinensis* (6156 inds. and 206.01 g/m$^2$), the highest biomass of *M. chinensis* was found in grade II (1024 inds. and 1876.14 g/m$^2$), and the occurrence of this species was centered in grades II and III (frequency = 50%), which is far from the polluted state. This misclassification seemed to be largely caused by the dominance of small-sized non-polychaete species, as reported previously [70]. Some minor exceptions between the ISEP grades and the conceptual P-R model should not be major concerns. As Dauvin [8] argued, no single index provides a full spectrum of environmental quality per area or region. Thus, these differences should be considered as a need for careful handling of an index, guided

by expert judgments, for local/regional calibrations [13], or the need to adopt a multimetric approach in environmental assessments after careful evaluations.

The question of whether the SAB curves and taxonomic variation along the ISEP grades in this study correspond to those of the P-R model has been addressed in many aspects. We found a high agreement in the graphical depictions of taxonomic composition between the ISEP and P-R models. One minor difference is that we observed high biomass in normal status samples, based on the ISEP grades. Echinoderms, mollusks, and arthropods appeared to be responsible for this phenomenon, which is discussed Section 4.3.

### 4.3. Compatibility with WFD Habitat Classification

The European Union Water Framework Directive (EU WFD) provides five habitat classification categories: high, good, moderate, poor, and bad. Because different indices use different systems for classifying environmental status, adopting WFD as the standard classification system for better communication has been encouraged [71,72]. The high correspondence between the ISEP and other established European indices that have already adopted the WFD classification system in terms of successional changes would motivate us to align the ISEP grades to the WFD categories. Thus, in this section, we summarize comparisons with other indices and designate the ISEP grades in terms of the WFD categories.

For the benthic habitat quality (BHQ) index proposed by Nilsson and Rosenberg [73], the habitat characterized by the PO was stage 1, which was designated as poor status in a later paper [46]. Based on this, the maximum abundance observed in Figure 4 allowed us to designate ISEP grade VI as a poor status. No grade was assigned to the ecotone point, although it was clearly identified between ISEP grades VI and V (Figure 4). The ecotone point was used to divide Stages 1 and 2 of Nilsson and Rosenberg [73], later assigned to a moderate status in Rosenberg et al. [46]. The grade was not assigned to the ecotone point in the ISEP, probably because the number of samples representing the characteristics of the ecotone point in the sample used for the percentile-based grade setting of ISEP was insufficient. Again, this may be because the samples of ecotone points where communities of two different status types meet have intermediate characteristics that can be easily changed to either. Nilsson and Rosenberg [73] divided status based on the ecotone point, which can also be explained through this reasoning.

The next successional stage set by another characteristic in which benthic enrichment appeared was the transitory phase (TZ) in the P-R model. In Figure 4, the shapes of the species and abundance curves in ISEP grades V and IV are very similar to those in this phase. In contrast, the biomass curve of the ISEP was slightly different from the P-R model, in which the secondary biomass maximum (sensu Pearson and Rosenberg [6]) was absent. Biomass in grade V was similar to the ecotone point in that a slight depression appeared after PO (Figure 4). This absence would be due to the great inherent variability in biomass, as noted by Pearson and Rosenberg [6]. Considering the lack of grade assignment on ecotone point and biomass in ISEP grade V sharing similarity with ecotone point, we divided the transitory phase into grades V and IV and regarded it as compatible with moderate and good quality status. This was the same case for the moderate status division of BQI that included enrichment in the transitory phase in part [46].

The SAB response curves along the ISEP grades fit well with those of the P-R model between grades VII (bad) and III (high). However, from grade II, the biomass increased, and the average biomass in grade I ($454 \pm 379$ gWWt/m$^2$) was five times higher than the overall mean of the study area ($92.5 \pm 215.2$ gWWt/m$^2$; Table 3). This was an unexpected outcome because the other two biological parameters, species number and abundance, still followed the P-R model for grades I and II. This departure can mainly be attributed to the biomass contribution of the heart urchin, *Echinocardium cordatum*. The spatangoid urchins, *E. cordatum* and *Schizaster lacunosus*, were the dominant species in terms of biomass, with higher occurrence frequencies in this study (frequency = 22.5%). Spatangoid urchins are

key sediment bioturbators that enhance ecosystem performance by improving nutrient cycling and the conditions for primary production [74].

Odum [75] suggested a conceptual model of ecological succession that could be applied across very different habitats, including marine ecosystems, and in the face of disturbances from widely varying sources and nature [76]. According to the Odum model, biomass monotonically increases from developmental stages, with a maximum in a stabilized, mature system. This pattern was similar to that of the biomass variation along the ISEP grades. Sea urchin habitats, which convey a higher biomass and better functional value on ISEP grades I and II, are susceptible to and impacted by anthropogenic disturbance activities, such as bottom trawling and dredging [74,77,78]. This habitat requires long-term recovery when disturbed because the life span of sea urchins occasionally exceeds 100 years [79]. Therefore, the presence of this habitat may reflect the absence of physical disturbances over long periods. While the biomass level in grade III undoubtedly indicates a healthy/normal condition, grades I and II may help identify untouched, mature communities. We expect that such a reflection of the biomass response in the ISEP will also be useful in assessing other physical disturbances (e.g., scouring in offshore wind farms).

We previously reported that the ISEP grade was independent of species diversity. Nevertheless, another interpretation is possible when we explore the section-by-section relationship among the ISEP grades. The ISEP is simply a function of species and biomass diversity. The former increases with the number of species, whereas the latter decreases with higher biomass. It is likely that the change in the ISEP grades from GPZ (VII) to TZ (IV and V), as shown in Figure 4, was caused by the species number. Subsequently, the grade change was driven by biomass (i.e., a decrease in biomass diversity in the denominator of the ISEP formula). The biomass curve followed Odum's succession model (max. biomass at the mature stage). However, the species number should have increased linearly if it followed the prediction by the Odum model. Odum [75] suggested that whether species diversity increases during succession depends on whether the increase in potential niches resulting from increased biomass exceeds the countereffects of increasing size and competition. One possible explanation for the decline in species diversity in grades I and II is competitive exclusion, which explains the decrease in species richness in a stable environment after removing disturbances [80,81]. This competitive exclusion theory may support the idea that ISEP grade I is indicative of a mature ecosystem.

The seas surrounding Korea rank among the highest levels of normalized biodiversity (i.e., species richness per unit area) worldwide [4]. As previously described, the Yellow Sea is one of the most highly productive ecosystems on earth (class 1, >300 g C/m$^2$/year). Despite the specificity in ecosystem characteristics, the results undoubtedly suggest that the patterns of SAB along the ISEP grades, of which the grade criteria were set by percentile scores of the ISEP, were highly similar to the P-R model. It also suggests that the ISEP would be a very useful and informative tool for assessing environmental quality in this region. Notably, the comparisons confirmed the similarity and compatibility of the ISEP with the WFD habitat classification system, which was developed under the guidelines of the EU WFD. This indicates the ISEP's potential generality and applicability to other coastal habitats after local/regional characteristics are properly incorporated and calibrated.

## 5. Conclusions

(1) We validated the performance of a macrobenthos-based ecological index, the ISEP, in the west coast of Korea. The validation was performed by examination of the linear relationship between the ISEP grades and environmental factors, and the correspondence of species-abundance-biomass (SAB) and taxonomical variation between the ISEP and the Pearson-Rosenberg (P-R) model.

(2) The ISEP responded effectively to bottom-water SS, a key impact source in this area, but was found to be independent of natural habitat condition variability such as salinity and sediment mean grain size. The SAB curves and the variation of major

taxa and dominant species along the ISEP grades corresponded well with those of the P-R model.

(3)     Conformity to the P-R model confirmed the compatibility of the ISEP index with the European Union's Water Framework Directive (EU WFD) classification system of ecological status and indicated the potential applicability to other areas.

**Author Contributions:** Data curation, formal analysis, writing—original draft, J.-W.Y. and Y.-W.L.; investigation, M.-R.P., C.-S.K., S.K., C.-L.L. and S.-Y.J.; conceptualization, J.-W.Y., D.L. and S.-Y.O.; methodology, J.-W.Y. and S.-Y.O.; writing—review and editing, J.-W.Y. and S.-Y.O.; funding acquisition, S.-Y.O. All authors have read and agreed to the published version of the manuscript.

**Funding:** We thank the relevant staff at the National Institute of Fisheries Science and the Korea Marine Environment Management Corporation for providing data. This study was conducted under the research project "Environmental Impact Analysis on the Offshore Wind Farm and Database System Development", which was driven by the Korea Environment Institute (KEI) and funded by the Korea Institute of Energy Technology Evaluation and Planning (KETEP) and the Ministry of Trade, Industry and Energy (MOTIE) of Korea (Project No. 20203030020080, PN90910).

**Institutional Review Board Statement:** This research was conducted in compliance with the guidelines of the Institutional Animal Care and Experimental Committee of the Korea Institute of Ocean Science and Technology (KIOST).

**Informed Consent Statement:** Not applicable.

**Data Availability Statement:** All data generated and/or analyzed during the current study are available from the corresponding author on reasonable request.

**Conflicts of Interest:** The authors declare no conflict of interest.

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
