# Peer review of "Application and Validation of an Ecological Quality Index, ISEP, in the Yellow Sea"

_jmse, doi:10.3390/jmse10121908_

Round 1

Reviewer 1 Report

This work provide a new index for ecological quality assessment, ISEP by test its application and validation in the Yellow Sea. Considersing the performance of ISEP in the realtionship with environmental factors and P-R model and taxonomical variation, the ISEP was verified to be a useful index tool to assess the local ecological qualtiy. Besides, the present work is well-orgnasized and well writen. So, I suggest it is accepted after minor revision.

Four questions I concerned:

1. The generality of this new biolodial index. Is it only applicable for the local study area with special heavy bottom water SS or other coastal water zone?

2. The author explain the reason of the ISEP with no linear relationships with some natural environmental variables, especially the depth, temperature, salinity and sediment textural parameters, which are all very crucial for macrobenthos living. So, Why? explain more.

3. There are many indices have been proved to be effective, such as AMBI, M-AMBI. I suggest the author to compare ISEP with M-AMBI to see if there are difference in performance.

4. So, is the ISEP independent of species diversity or not?

Reviewer 2 Report

The reviewed ms provides a validation of the ISEP index used in Korean coastal monitoring system. It falls within the scope of the journal as well as the Special Issue. Its major advantage is the availability of an excellent dataset with a large number of stations and many environmental variables. Also the authors have a deep understanding of benthic ecology as well as the relative scientific literature. My concerns focus on two points:

1) No data is available to the reader. Data availability is important and enables the reviewer and reader to verify the results presented. Furthermore data availability would generate significant citations to the ms benefiting both the journal and the authors. I suggest to upload the data to OBIS.

2) Some important sections in the M&M are quite confusing. The methodology is sound but there is a need for simpler presentation of those methods.

Due to the large sections that should be reviewed my suggestion is “major revision”

Introduction

L41: Change term heavily “used”

L48: Initiate and improve

L58, Ref [7]: Please find a more recent citation

L63: I think that the study of Choi, Jin-Woo & Seo, Jin-Young (2007). Application of biotic indices to assess the health condition of benthic community in Masan Bay, Korea DOI: 10.4217/OPR.2007.29.4.339 should also be mentioned in the introduction and the findings compared at the discussion section since it is also a S. Korean study.

Please describe the current legislation system for benthic monitoring in S. Korea in a small paragraph. It there is none please mention how your study could contribute to its development

Materials

Section 2.1 The authors provide a detailed description of the study region with different parameters. It would be nice to have a colored map at this section presenting all those details. I understand that this is figure may be complicated to create so please treat this comment as an optional suggestion.

L152: The sampling was performed using a research vessel?

L151: Please explain seasonal intervals. Also please explain more when the samples where collected. If I understand correctly different stations were sampled each year. This should be treated with caution, was at least, the sampling season the same across all samplings? T

L159: Sampling season is even more important since water was sampled and used as an environmental parameter.

L161-163: This phrase is a bit confusing. Surface and bottom water was sampled. Subsamples from each depth where collected and analyzed. Averages were calculated at each depth or from the whole water column (combining the 2 depths).

L166: Similar comment here

L169: Water was noy filtered first? Also need a reference here 

L172: The quality of the image should be improved and color to be added. As it is, readers may confuse sampling stations with small islands.

         Also is it possible to draw the border between South Κorean and North Korea? Sinse the proposed index is only valid there…

L175: I think the term “where macrofauna was sampled” is more widely used.

L178-204: How were the sediment samples collected? Using a corer from within the grab? From how many cm deep in the sediment was each same collected? Was is the same for all parameters? Where there any replicates like macrofauna?

Redox potential is a common env. parameter usual highly correlated with benthic macrofauna and indices. Was it not measured?

L226: averaged value? Instead of score.

L227: Pearson correlation is a parametric method and requires normality, which in my experience is seldom the case on such data. Please check for normality or use Spearman rank correlation instead.

L230-232: This sentence should be deleted or moved in the discussion section.

L234: again the term grade is used. Since the ISEP output is a number another term should be used.

L233: “variations” in SAB. I think that using the term variance implies that some sort of analysis was made. Please explain this part more. d

L232-240: I am sorry but I cannot understand how this analysis was made… If it is explained more in section 2.4 please remove this part from here.

L241: This is the description of the third validation?

L241-247: Again this method is not clear and as it is presented I am not sure it is appropriate as a method of validation. What are the “successive taxonomical changes originally proposed in the P-R model” ? What there a statistical method involved in the comparison? How did you know the k/r strategy of the selected species?

L248-257: This is a sound approach by I would like more support to the rationale behind the separation of the P-Rν model. For example why the cutoff point between part 1 – 2 and 2- 3 was decided? Why set group 2 at equal points of the plot or at the same level on the ascending and descending points on the curve?

L300: More data is needed to be presented and made available to the reader. A summary of environmental variables is not enough. Also no biotic data is available nor the values of the IESEP index. In my opinion it is necessary to upload the all data to a public repository like OBIS. The validation provided by the OBIS as well as the data availability to the reader would greatly benefit the publication. If the authors do not prefer to make abundance data available they should at least provide the data as presence/absence along with coordinates, environmental variables and ISEP values for each station.

L323: I agree but this comment should be placed to the discussion section in the appropriate section

L386: Again they authors should provide more support as to how some species were selected and other excluded.

L439: TOC is, in my experience, an excellent indicator of organic enrichment disturbance. More analysis should be made to explain why it was not correlated to ISEP. Perhaps separating the dataset in disturbed – undisturbed conditions and looking at the lower end? The paper bellow may held.

Simboura, N., Zenetos, A. & Pancucci-Papadopoulou, M.A. Benthic community indicators over a long period of monitoring (2000–2012) of the Saronikos Gulf, Greece, Eastern Mediterranean. Environ Monit Assess 186, 3809–3821 (2014). https://doi.org/10.1007/s10661-014-3659-z

Discussion: The authors provide a long and informative discussion with plenty of supporting references. However some sections may be better placed in the M&M to help the reader understand some choices made by the authors. I will not provide more comments for this section at this point. However I would suggest to shorten the section and make it more “to the point”. For example I believe that the paragraph in L545-557 is not very relevant as the BHQ was not calculated in this study…

Reviewer 3 Report

It is rare that I can find very little to comment upon in a submitted manuscript. This manuscript is very well written and I have no substantive technical comments. My comments are primarily about the color choices used in some of the graphics. While these issues may be the result of my printer, I don't think so. 

Figure 3: The use of light green is difficult to see. Using another color or just B&W would be preferable.

Figure 5: Using light green from the scale along the x-axis is difficult to see. Using just black would be preferable.

Figure 7: Using light green for the axes is difficult to see. Using black would be preferable.

Figure 8: Using light green for the axes is difficult to see. Using black would be preferable.

Round 2

Reviewer 2 Report

The authors have addressed or explained all points raised. 

Author Response

Point 1. The authors have addressed or explained all points raised. 

Response 1. We appreciate the reviewers' comments.